# Forest Fire Object Detection Analysis Based on Knowledge Distillation

**Jinzhou Xie and Hongmin Zhao ***

College of Computer & Information Engineering, Central South University of Forestry and Technology, Changsha 410004, China; 20221100409@csuft.edu.cn
* Correspondence: t20020596@csuft.edu.cn

**Abstract:** This paper investigates the application of the YOLOv7 object detection model combined with knowledge distillation techniques in forest fire detection. As an advanced object detection model, YOLOv7 boasts efficient real-time detection capabilities. However, its performance may be constrained in resource-limited environments. To address this challenge, this research proposes a novel approach: considering that deep neural networks undergo multi-layer mapping from the input to the output space, we define the knowledge propagation between layers by evaluating the dot product of features extracted from two different layers. To this end, we utilize the Flow of Solution Procedure (FSP) matrix based on the Gram matrix and redesign the distillation loss using the Pearson correlation coefficient, presenting a new knowledge distillation method termed ILKDG (Intermediate Layer Knowledge Distillation with Gram Matrix-based Feature Flow). Compared with the classical knowledge distillation algorithm, KD, ILKDG achieved a significant performance improvement on a self-created forest fire detection dataset. Specifically, without altering the student network's parameters or network layers, mAP@0.5 improved by 2.9%, and mAP@0.5:0.95 increased by 2.7%. These results indicate that the proposed ILKDG method effectively enhances the accuracy and performance of forest fire detection without introducing additional parameters. The ILKDG method, based on the Gram matrix and Pearson correlation coefficient, presents a novel knowledge distillation approach, providing a fresh avenue for future research. Researchers can further optimize and refine this method to achieve superior results in fire detection.

**Keywords:** knowledge distillation; forest fire detection; YOLOv7; YOLOv7x; object detection





## 1. Introduction

Forest fires [1] are a destructive natural disaster that severely impacts ecosystems and human societies [2]. With the increasing frequency and intensity of forest fires due to global climate change and human activities, they have become a significant challenge for the current ecological environment and social safety. Early detection and rapid response to forest fires are crucial for minimizing disaster losses, and the application of object detection technology provides new possibilities to enhance fire monitoring and response efficiency. Object detection technology [3–5] is a crucial task in computer vision, aiming to detect and identify specific objects from complex scenes. In forest fire monitoring [6,7], object detection can assist in automatically identifying signs of fires, such as smoke and flames, enabling early detection and localization of fires. Through in-depth research on the application of object detection in forest fire monitoring, we hope to enhance disaster response capabilities and reduce losses caused by fires. Many aspects of object detection methods [8,9] still need to improve in forest fire detection, such as high network complexity, weak generalization ability, difficulty deploying large models, and insufficient data samples affecting model accuracy. Lu et al. proposed a forest fire detection model based on multi-task learning [10], including three tasks (detection, segmentation, classification) and one shared feature extraction module. Although it improved performance, the model's

complexity remains high. Zhao [11] proposed the Fire-YOLO deep learning algorithm, expanding the network from three-dimensional features to enhance performance, but its model robustness still needs to be validated. Wang et al. [12] introduced a novel developed channel attention module to improve detection performance, but it also involves more computational costs. Gong et al. [13] proposed a flame centroid stabilization algorithm based on spatiotemporal relationships. This algorithm calculates the centroid of the flame region in each image frame, incorporating temporal information to obtain spatiotemporal details of the flame centroid. However, it is worth noting that the application scenarios for the proposed algorithm are limited. This paper combines object detection technology with knowledge distillation to address the challenges in forest fire detection models, such as high complexity, weak generalization ability, and deployment difficulty. This approach seeks to drive the optimization and progress of forest fire detection models.

Knowledge distillation is a methodology that aids in training a smaller network (student) guided by a more extensive network (teacher). Generally, large models manifest as either a singular intricate network or an assemblage of multiple networks, demonstrating robust performance and generalization abilities; large models contrast with small models that exhibit constrained expressive capacity owing to their compact network size [14,15]. Consequently, the knowledge obtained from complex and powerful models can guide the training of compact and efficient models, enabling them to achieve performance comparable to that of large models but with a significantly reduced number of parameters. This approach facilitates model compression and acceleration [16,17]. Hinton et al. [18] first introduced a technique called Knowledge Distillation (KD), which utilizes a teacher network's softmax outputs for instructing a student network's learning. FitNets [19] proposed a hint-based approach, aligning the intermediate layers of the student network with those of the teacher network. During the distillation of information (in the middle layers), because of the mismatch in feature dimensions of the teacher and student networks, dimensionality reduction is used to match the dimensions, thereby transferring knowledge. Still, this reduction also leads to some information loss [20,21]. In recent methods (FT [22], AB [23]), the aim is to increase the amount of information transmitted during the distillation process to enhance distillation performance. Both techniques exhibit improved distillation performance through the augmentation of transferred information. However, FT and AB alter the teacher's feature values, leaving room for further performance improvement. The primary purpose of this study is to cleverly integrate YOLOv7 [24] with knowledge distillation techniques and propose ILKDG to enhance the performance of forest fire detection [25] while keeping network parameters and layers unchanged. This study explores an approach based on Gram matrices to describe the problem-solving process. Introducing a computation method for cross-layer Gram matrices sets this research apart from previous studies [26].

In contrast, prior research mainly focused on computing dot products between intra-layer features. To more effectively transmit knowledge, we incorporate the Pearson correlation coefficient [27,28] and redesign the loss function to optimize the training [29] of the student deep neural network (DNN) to align its FSP matrix with that of the teacher network. Experimental results demonstrate that our approach achieves significant performance improvements on our self-created forest fire detection dataset. Compared with the traditional knowledge distillation approach, while keeping the number of parameters and network layers unchanged, our ILKDG method increases mAP@0.5 by 2.9%, while mAP@0.5:0.95 saw a 2.7% improvement. Precision and Recall also show significant progress, which are encouraging results. Furthermore, compared with widely adopted object detection distillation methods, our ILKDG method exhibits strong competitiveness, offering a new avenue for performance enhancement in forest fire detection.

## 2. Background and Related Work

### *YOLOv7 Architecture*

YOLOv7 represents the most recent advancement in the YOLO series, further enhancing detection speed and accuracy based on previous iterations. The above paper introduces E-ELAN in the overall architecture, utilizing shuffle, expand, and merge cardinality to consistently bolster the network's learning capabilities without compromising the initial gradient path. E-ELAN enables the guidance of different groups of computational blocks to acquire diverse features. The authors also propose a compound model scaling technique to retain the properties of the initial design and uphold the optimal structure. Concerning network optimization strategy, these authors introduce and refine model re-parameterization and dynamic label assignment, addressing their existing challenges.

Regarding the former, the authors note that the identity connection in RepConv [30] provides more gradient diversity for different characteristic graphs by directly accessing the cascade of ResNet [31] or DenseNet [32], potentially disrupting the network structure. Consequently, the authors eliminate the identity connection in RepConv and devise a planned reparameterized convolution, efficiently combining re-parameterized convolution with various networks. For the latter, their paper incorporates the concept of Deep Supervision [33]. It introduces an additional head in the intermediate layer of the network, serving as an auxiliary loss to instruct the learning of the weights of shallow networks. A novel label assignment method is formulated for this structure. Figure 1 shows the YOLOv7 framework.

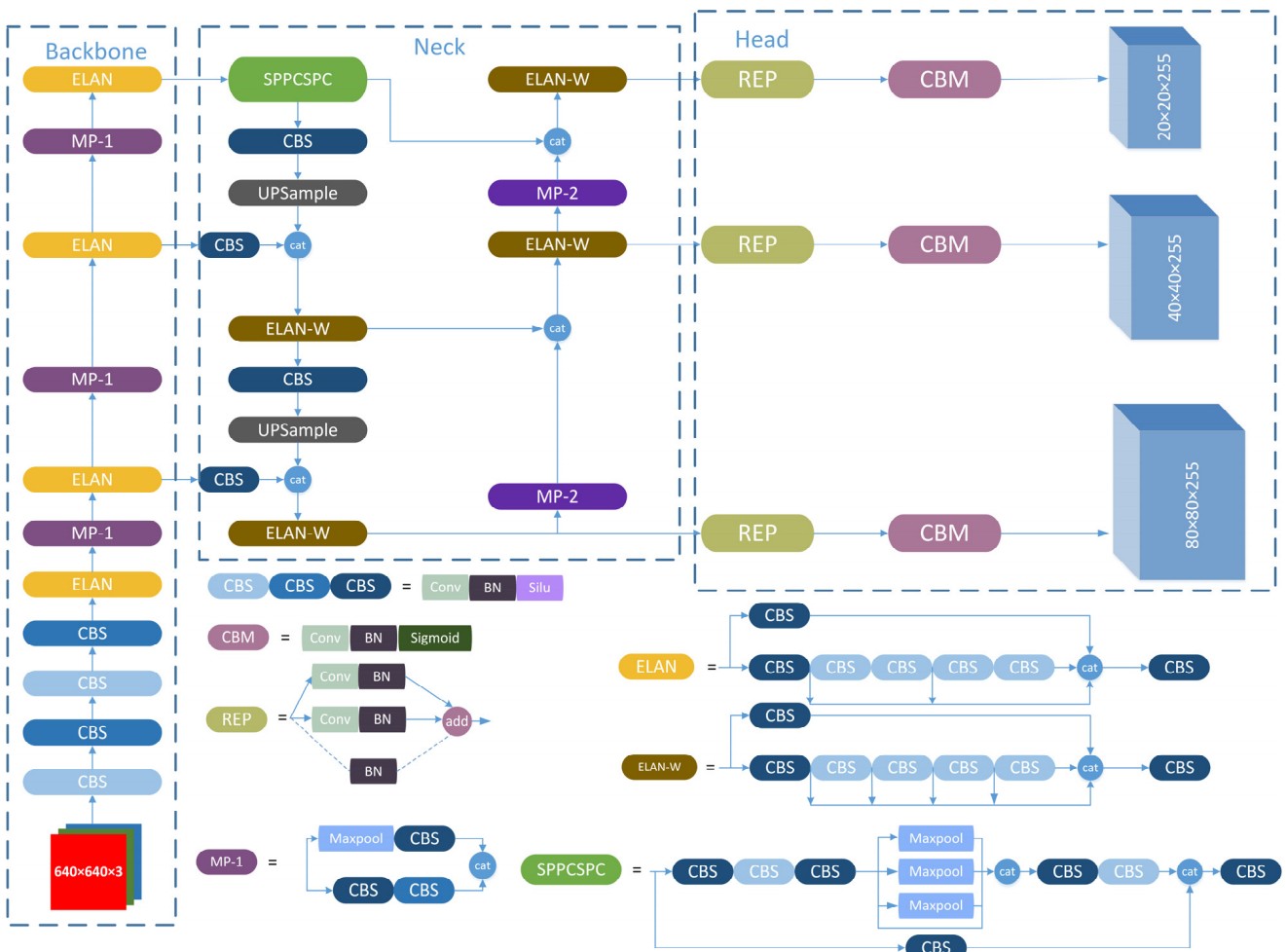

**Figure 1.** The network architecture of YOLOv7.

To perform knowledge distillation, we selected the YOLOv7x network, which is structurally similar to YOLOv7, as the teacher network to ensure compatibility between the two models and improve model performance. The complexity of the teacher network typically surpasses that found in the student network, but the difference should be manageable to avoid poor knowledge distillation results. The basic parameters of the two models are summarized in Table 1.

**Table 1.** Comparison between YOLOv7 and YOLOv7x.

| Model | Layers | Parameters | GFLOPS | Size (MB) |
|---|---|---|---|---|
| YOLOv7 | 415 | 31189962 | 93.7 | 71.8 |
| YOLOv7x | 467 | 73147394 | 190.61 | 142.1 |

## 3. Materials and Methods

### 3.1. Knowledge Distillation

Knowledge distillation [34] involves the process of transferring knowledge from a sizable, intricate model (teacher model) to a more compact, effective model (student model) [35]. It involves training the student network to mimic the teacher network's output and emulate its internal representations or decision-making process. This technique is used to enhance the performance of smaller models, making them approximate the behavior of larger models while reducing computational costs and memory requirements. In the process of knowledge distillation, logits are used as a basis for comparing the outputs of the student model with those of the teacher model. By examining logits, the model can measure the certainty or confidence of its predictions before applying the softmax function to obtain the final probability. The standard cross-entropy loss depends on the predicted probabilities and ground truth labels. Additionally, the loss function is extended to include the standard cross-entropy loss between the predictions of the student model and the ground truth labels, as well as an additional loss term that measures the discrepancy between the softened probabilities (obtained through a higher temperature softmax) of the teacher model's predictions and the corresponding predictions of the student model. This additional term ensures that the student model not only learns to predict the correct labels but also aims to replicate the softened outputs of the teacher model, effectively transferring its knowledge to the student model. By jointly minimizing these two loss terms, the student model can learn to generalize better and imitate the behavior of the more complex teacher model. The calculation of the probability for the class is as follows:

$$p_i = \frac{exp\left(\frac{z_i}{T}\right)}{\sum_j exp\left(\frac{z_j}{T}\right)} \tag{1}$$

Here, $T$ represents the "temperature" of knowledge distillation. When $T = 1$, it corresponds to a normalized exponential function. With the increase in the temperature parameter $T$, the softmax function's probability distribution becomes smoother, thereby conveying more nuanced particulars about the interrelation of different categories according to the teacher model. This information, referred to as "dark knowledge" by Hinton, is what we aim to impart to the student model in distillation. To compute the loss function for the teacher's soft targets, we use the same $T$ value to calculate the softmax function based on the student logits. This kind of loss is frequently called "distillation loss." Therefore, with the increase in $T$, we are better able to impart the knowledge of the teacher model to the student model, aiding the latter in learning and generalization.

In 2015, Hinton et al. discovered the advantages of training the distilled model to not only produce the soft labels from the teacher but also the correct labels relying on the ground truth labels. Consequently, we compute the "standard" loss by comparing the predicted class probabilities of the student with the ground truth labels. This loss is named the "student loss". When calculating the class probabilities for the student loss, we employ

$T = 1$. The comprehensive loss function, which integrates both the distillation and student losses, is determined as follows:

$$L(x; W) = \alpha \times H(y, \sigma(z_s; T = 1)) + \beta \times H(\sigma(z_t; T = \tau), \sigma(z_s, T = \tau)) \tag{2}$$

Here, $x$ represents the input, $W$ represents the student model's parameters, $y$ denotes the ground truth labels, $H$ signifies the cross-entropy loss function, $\sigma$ represents the softmax function characterized by the "temperature" $T$, and $\alpha$ and $\beta$ are constants. The logits of the student and teacher are denoted as $z_s$ and $z_t$, respectively. The general structure of knowledge distillation is depicted in Figure 2.

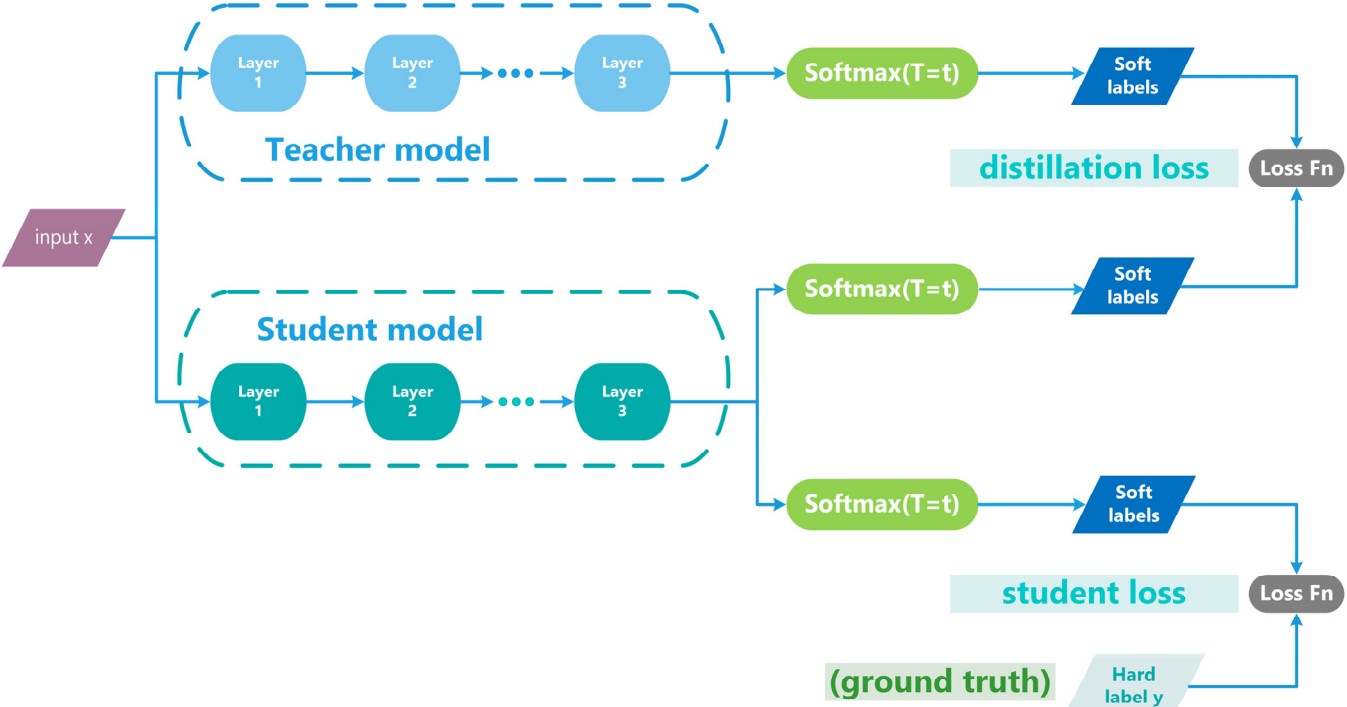

**Figure 2.** A generic framework for knowledge distillation.

### 3.2. Flow of Solution Procedure (FSP) Matrix

Deep Neural Networks progressively generate features [36–38], with higher layers capturing more closely related critical features necessary for the main task. Considering the training process of a DNN as the problem and its learned weights and parameters as the solution, the features generated within the depths of the DNN can be viewed as temporary results during the solving process. Based on previous experience, Romero et al. proposed knowledge transfer techniques, enabling a student network to study the intermediate layer outputs of a teacher network. However, numerous approaches exist in the context of DNNs for solving the mapping problem from input to output. Consequently, replicating the features generated by the teacher's DNN can impose rigid constraints on the student's DNN.

In human cognition, teachers typically elucidate the process of arriving at a solution when addressing challenges. In contrast, students engage in learning the problem-solving procedure. In specific problem scenarios, a student Deep Neural Network (DNN) may not necessarily require the acquisition of intermediary outputs for a given input; instead, it can assimilate the methodologies for tackling these particular problem types. Through this approach, mastering the problem-solving process can offer more substantial assistance than merely instructing intermediate outcomes.

The effectiveness of knowledge transfer [39–41] depends significantly on the precise definition of refined knowledge. The extracted knowledge is usually obtained through

different features in pre-trained DNNs. Drawing an analogy from how a skilled teacher imparts the problem-solving process to a student, we define highly refined knowledge as the problem-solving procedure. Given that neural networks process input data to yield corresponding output results and a mapping relationship exists between the inputs and outputs, we can understand the problem-solving process as the connection among features at different layers of the neural network.

The Gram matrix, generated by calculating the dot products between feature vectors, inherently captures the directional characteristics among features. Inspired by this, Gatys et al. represented the pattern details of input data by computing the Gram matrix within layers. In line with their work, our approach utilizes the Gram matrix, which is composed of the dot products of different features derived from two layers in the neural network, to symbolize the problem-solving process. Our method's primary distinction lies in utilizing features from different layers to calculate the Gram matrix. In contrast, the Gram matrix in [26] is derived from deducing dot products within layers. Based on this, we use different features from two layers to construct the matrix. Afterward, the student model learns to align its matrix with the teacher model's. As shown in Figure 3, a method for transferring refined knowledge is presented by generating a Flow of Solution Procedure (FSP) matrix of the problem-solving process from the extracted features of two layers.

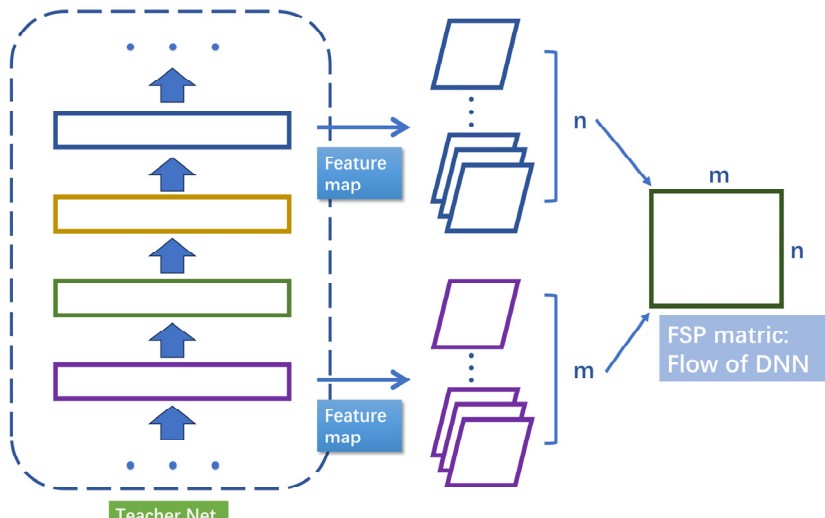

**Figure 3.** Schematic diagram of FSP. Two layers make the Flow of Solution Procedure matrix of features and represent the process or solution through which the teacher network solves problems.

The interplay between two intermediary outcomes can characterize the progression of the solution procedure. In the context of DNNs, we can consider the directional relationship between features from different layers from a mathematical perspective. We have introduced the *FSP* matrix as a means to depict this progression. The *FSP* matrix, denoted as $G \in \mathbb{R}^{m \times n}$, is generated from the features of two selected layers. We consider one layer producing feature maps $F^1 \in \mathbb{R}^{h \times w \times m}$, where $H$, $W$, and $M$ correspond the height, width, and number of channels, respectively. Moreover, another feature map is defined as $F^2 \in \mathbb{R}^{h \times w \times m}$. Then, we compute the *FSP* matrix as follows:

$$G_{i,j}(x; W) = \sum_{s=1}^{h} \sum_{t=1}^{w} \frac{F^1_{s,t,i}(x; W) \times F^2_{s,t,j}(x; W)}{h \times w} \tag{3}$$

Here, $x$ represents the input data, while $W$ represents the weight parameters of the neural network. Multiple points have been selected to create the Flow of Solution Procedure matrix, as illustrated in Figure 4.

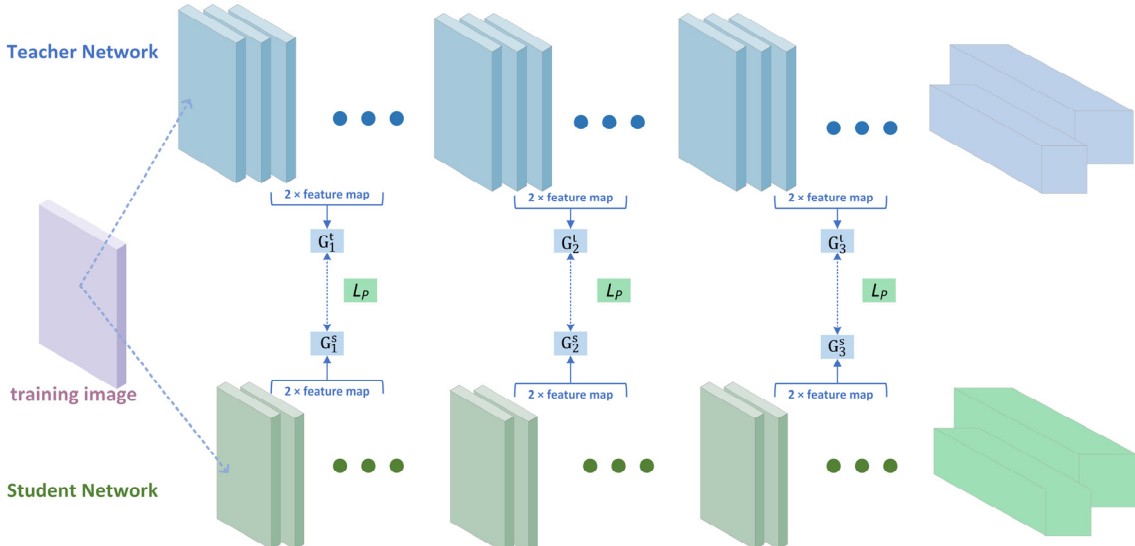

**Figure 4.** Overall framework diagram of ILKDG. The number of layers in the student and teacher models can be adjusted based on practical considerations, and the required quantity of FSP matrices (in the same spatial size) may vary for different tasks.

### 3.3. Loss Function

The Pearson correlation coefficient a statistical measure used to assess the linear connection between two variables. When evaluating the connection between teacher and student networks, using the Pearson correlation coefficient can help us understand the degree of similarity in their feature generation and prediction [42,43] results. Specifically, when the outputs of two networks exhibit similar variations on similar inputs, their Pearson correlation coefficient will be closer to 1, indicating a strong positive correlation between them. Conversely, if their output trends are opposite, the correlation coefficient will be closer to $-1$, indicating a strong negative correlation. When their output variations do not exhibit a linear relationship, the correlation coefficient will approach 0, suggesting a low correlation between them.

When evaluating the connection between the teacher and student models, the Pearson correlation coefficient can help us evaluate the effectiveness of techniques like knowledge distillation. Suppose the student network can partially imitate the output patterns of the teacher network through knowledge distillation. In that case, their Pearson correlation coefficient may be relatively high, indicating that the student model can effectively acquire knowledge from the teacher model. In summary, the Pearson correlation coefficient quantitatively measures the association between teacher and student networks, aiding in the assessment of knowledge transfer and learning effectiveness. By estimating the sample covariance and standard deviations [44], one can obtain the sample correlation coefficient (sample Pearson coefficient), $\rho\,(u,v)$, which signifies the Pearson correlation coefficient between two random variables, $u$ and $v$.

$$\rho(u,v) := \frac{Cov(\boldsymbol{u},\boldsymbol{v})}{Std(\boldsymbol{u})Std(\boldsymbol{v})} = \frac{\sum_{i=1}^{C}(u_i - \bar{u})(v_i - \bar{v})}{\sqrt{\sum_{i=1}^{C}\left(u_i - \bar{u}\right)^2 \sum_{i=1}^{C}(v_i - \bar{v})^2}} \tag{4}$$

where $Cov(u,v)$ is the covariance of $u$ and $v$, and $\bar{u}$ and $Std(u)$ represent the mean and standard derivation of $u$, respectively.

We then impart the knowledge extracted from the teacher network to the student network to enhance its performance. As previously discussed, we represent the extracted knowledge as FSP matrices, which depict the methodology or procedure employed by the teacher network in problem solving. Suppose the teacher network generates n FSP

matrices, denoted as $G_i^T$, where $i = 1, \ldots, n$. Simultaneously, the student network also generates n FSP matrices represented as $G_i^S$ where $i = 1, \ldots, n$. In this study, we exclusively focus on Flow of Solution Procedure matrices $\left(G_i^T, G_i^S\right)$, where $i = 1, \ldots, n$, which possess identical spatial dimensions. We redefine the distillation loss by incorporating the Pearson correlation coefficient mentioned above. Therefore, the cost function for the knowledge transfer distillation task is defined as:

$$L_P(W_t, W_s) = \frac{1}{N} \sum_x \sum_{i=1}^{n} \lambda_i \left(1 - \rho \left(G_i^T(x; W_t), G_i^S(x; W_s)\right)\right)^2 \qquad (5)$$

Here, $\lambda_i$ and $N$ denote the weighting of each loss term and the total number of data points. In this study, it is assumed that all loss terms have equal importance. Thus, we use the same $\lambda_i$ in subsequent experiments.

The student network's acquisition of knowledge from the teacher network is primarily structured in two stages. First and foremost, we minimize the $L_P$ loss function to ensure the similarity of the Flow of Solution Procedure matrix between the student and teacher models. Secondly, given that we utilize a detection task to confirm the efficacy of the proposed approach, the student network undergoes further training in the second stage using task loss. Hence, we employ a weighted combination of classification *loss*, *objectness loss*, and *IOU loss* as the task loss. Therefore, the overall training loss, $L_{total}$, can consist of both the detection loss and the knowledge distillation loss, as follows:

$$L_{det} = \alpha \times lcls + \beta \times lobj + \gamma \times lbox \qquad (6)$$

$$L_{total} = \theta \times L_P + (1 - \theta)L_{det} \qquad (7)$$

where $\alpha$, $\beta$, $\gamma$, and $\theta$ take on different preset values depending on the specific task.

## 4. Experiments

### 4.1. Dataset

To enhance the model's generalization, mitigate overfitting and improve robustness against various variations and noise, this study employed the Mosaic and Mixup [30] data augmentation techniques. The data augmentation had the effects as shown in Figure 5. Post augmentation, the ultimate experimental dataset consisted of 3780 images. Subsequently, the dataset was divided into training set, test set, and validation set in a ratio of 6:2:2.

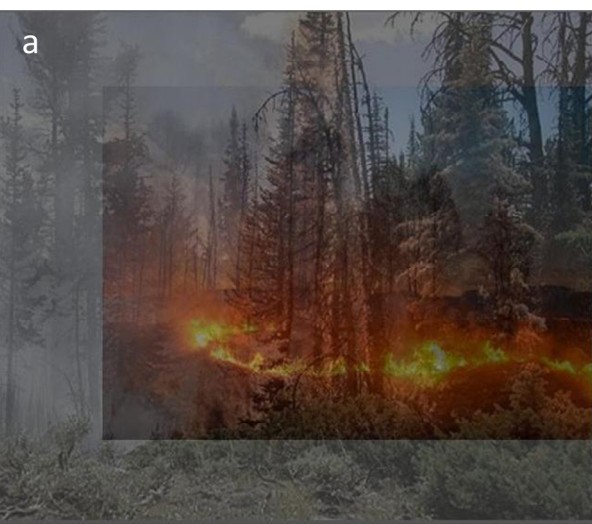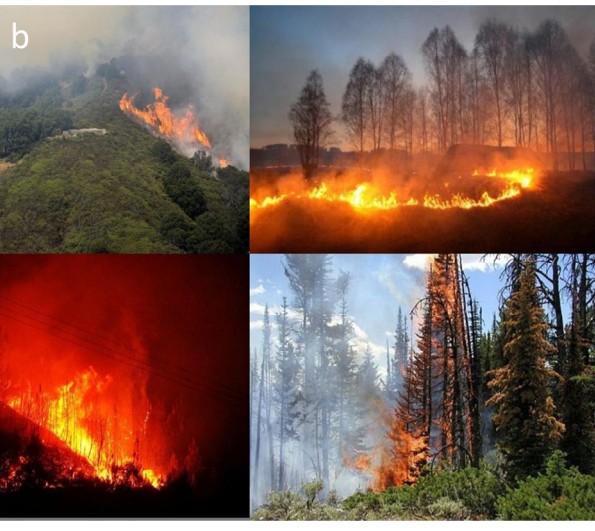

**Figure 5.** Examples of data augmentation. (**a**) demonstrates the Mixup data augmentation technique, while (**b**) illustrates the Mosaic data augmentation technique.

### 4.2. Experimental Configuration and Environment

Our experiment uses Python as the programming language and PyTorch as the deep learning framework on a system running Windows 11. Table 2 outlines the hardware specifications, while the training employed the following hyperparameters: an input image size of 640 × 640, 300 training epochs, and a batch size of 64. The model optimization settings can be found in Table 3.

**Table 2.** Experimental conditions.

| Experimental Environment | Details |
| --- | --- |
| Programming language | Python 3.8 |
| Operating system | Windows 11 |
| Deep learning framework | PyTorch 1.13.0 |
| GPU | NVIDIA GeForce GTX 3090 |
| GPU acceleration tool | CUDA:11.3 |

**Table 3.** Parameter Settings.

| Optimization Method | Initial Learning Rate | Momentum | Weight Decay |
| --- | --- | --- | --- |
| Stochastic Gradient Descent (SGD) | 0.01 | 0.973 | 0.0001 |

### 4.3. Evaluation of the Model

In this study, we assessed the quality of the model in two ways: recognition accuracy and lightness of the model. Therefore, we chose mAP@0.5 and mAP@0.5:0.95 as two metrics to assess the predictive accuracy of the model. In addition, we chose GFLOPs (Gigabit Floating Point Operations per Second) and model parameters as two metrics to measure the degree of model lightness.

(1) AP Metric: Within the confusion matrix, TP corresponds to the count of accurately predicted fire samples, while FN refers to the count of fire samples inaccurately predicted as non-fire. From these values, Precision (P) and Recall (R) can be derived, with P and R representing the precision and comprehensiveness of fire detection, respectively. The equations for computing P and R are presented in Formulas (8) and (9), respectively.

$$\text{Precision (P)} = \frac{\text{TP}}{\text{TP} + \text{FP}} \tag{8}$$

$$\text{Recall (R)} = \frac{\text{TP}}{\text{TP} + \text{FN}} \tag{9}$$

As shown in Formulas (10) and (11), they represent *mAP* and *IoU*, respectively.

$$mAP = \int_0^1 p(0)do \tag{10}$$

Here, $p(0)$ represents the attained level of accuracy in object detection. *IoU* is the ratio of the intersection area and union area of the predicted area and the actual area, which is usually used to measure the accuracy of the target detection algorithm [45].

$$IoU = \frac{Area_{pred} \cap Area_{gt}}{Area_{pred} \cup Area_{gt}} \tag{11}$$

(2) *GFLOPs* is a metric employed to quantify the time complexity of the model, which exhibits a direct relationship with the hardware performance demands. Equation (12) illustrates the formula utilized for computing *GFLOPs*.

$$GFLOPs = \left(2C_i K^2 - 1\right) HWC_0 \tag{12}$$

In the context of this formula, $C_i$ and $C_0$ denote the count of input and output channels, $K$ represents the kernel size, while $H$ and $W$ describe the dimensions of the feature map.

(3) Parameters refer to the quantity of parameters the model utilizes, typically measured in millions. This metric significantly impacts the ultimate size of the trained model's output.

### 4.4. Comparison of Experimental Results

The typical training process of distillation [46] generally comprises two main phases. Firstly, the teacher network is trained. Secondly, the student network is trained using the knowledge provided by the teacher network. Specifically, (1) YOLOv7x is trained as the teacher detector. (2) The knowledge transferred from the teacher network is defined, specifying intermediate layer feature representations and processing information as knowledge targets. (3) The loss function is defined. (4) The feature flow matrix representing the problem-solving process is incorporated into the loss function during the training process of the student model. (5) The student model is trained using the loss function. (6) After training, the accuracy of the student network is assessed using the test data to determine whether it successfully acquired knowledge from the teacher network while ensuring performance improvement.

In this study, we conducted object detection experiments using our self-created forest fire dataset and employed ILKDG technology. The experimental results illustrate a notable enhancement in the model's predictive capability following distillation, particularly in its ability to withstand adversarial noise and generalize effectively. Encouragingly, the distilled model exhibited a noticeable enhancement in the original object detection task, as shown in Figure 6, and successfully detected objects that were previously unidentifiable before distillation. This highlights the potential advantages of ILKDG technology in addressing adversarial noise and improving model generalization. These results emphasize ILKDG as a practical approach for enhancing object detection performance and provide valuable guidance for further research and applications.

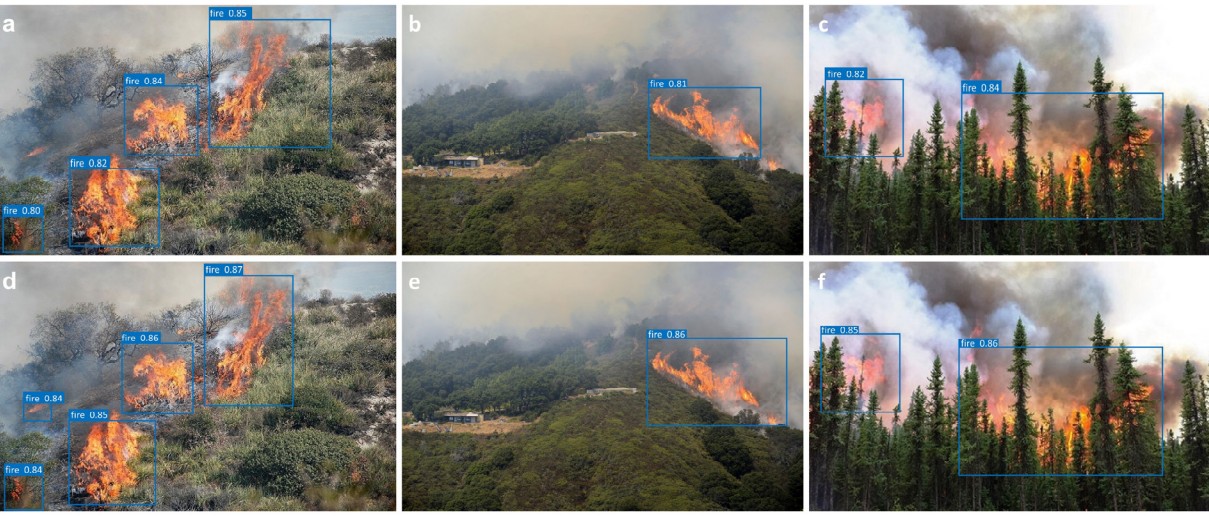

**Figure 6.** Performance comparison before and after distillation. (**a**–**c**) demonstrates the performance before distillation, while (**d**–**f**) illustrates the performance after distillation.

To assess the efficacy of the proposed approach, we conducted a comparative analysis of the self-created forest fire dataset compared with widely adopted object detection distillation methods. The results in Table 4 demonstrate significant improvements achieved by our approach. As expected, models with more parameters, or more extensive or deeper architectures exhibited better performance, while smaller or shallower models had advantages in terms of processing speed. First, we applied the KD (Knowledge Distillation) and FitNet methods and found that these techniques were effective in enhancing performance, leading to improvements in mAP@0.5 and mAP@0.5:0.95. Next, we contrasted the FGD [47] and KD++ [48] methods, which seemed to show no significant performance gains and, in some cases, resulted in performance degradation, particularly in terms of Recall.

**Table 4.** Evaluation results on self-created forest fire dataset.

| Model | Precision | Recall | mAP@0.5 | mAP@0.5:0.95 | Params (M) |
|---|---|---|---|---|---|
| Teacher (YOLOv7x) | 0.881 | 0.842 | 0.895 | 0.637 | 73.1 |
| Student (YOLOv7) | 0.833 | 0.819 | 0.846 | 0.604 | 31.1 |
| +KD | 0.844 | 0.824 | 0.859 | 0.618 | 31.1 |
| +FitNet | 0.847 | 0.822 | 0.860 | 0.611 | 31.1 |
| +FGD | 0.850 | 0.821 | 0.854 | 0.621 | 31.1 |
| +KD++ | 0.846 | 0.823 | 0.858 | 0.622 | 31.1 |
| +Proposed | 0.862 | 0.831 | 0.870 | 0.631 | 31.1 |

Finally, we employed our proposed ILKDG method, which outperformed all other methods in all metrics, especially in mAP@0.5 and mAP@0.5:0.95, where it achieved relative improvements of 2.9% and 2.7%, respectively, compared with KD. This performance gain was achieved while maintaining the same number of parameters (31.1 million). In object detection tasks, a 1% improvement is considered highly significant, especially on large-scale datasets.

These results highlight the significant potential for performance enhancement our proposed method offers on our self-created forest fire dataset, while maintaining a relatively low model complexity. Our research provides essential insights and performance improvements for wildfire detection in forests, improving the efficient surveillance and anticipation of natural calamities such as wildfires.

### 4.5. Ablation Study

To confirm the efficacy of our correlation-based approach, we measured the correlation between student and teacher models, with the student model being trained using standard classification loss, knowledge distillation (KD), and our ILKDG method. We chose commonly used metrics such as Pearson correlation coefficient, Spearman's [49], and Euclidean distance [50] as correlation measures. As summarized in Figure 7, ILKDG exhibited stronger correlations than the baseline.

We trained the student YOLOv7 and teacher YOLOv7x on the self-created forest fire dataset with or without label smoothing (LS). In our approach, we performed experiments to verify the effectiveness of the cosine similarity technique [51]. Both cosine similarity and Pearson correlation coefficient can be employed to evaluate the teacher's and student's association. Compared with the scale invariance of cosine similarity, the Pearson correlation has an additional shift-invariance that results from centering the vectors first, making it more robust to distribution changes. We performed experiments to compare the use of these two metrics in our ILKDG and train models with or without label smoothing. Recent research [52,53] indicates that knowledge distillation with high-temperature settings is incompatible with label smoothing, so we also employed KD ($\tau = 1$) to train our networks. ILKDG using Pearson correlation outperformed in terms of accuracy, especially when the teacher and student were trained with label smoothing (where predicted probability distributions may undergo shifts due to its influence), as indicated in Table 5. Therefore,

Pearson correlation, which possesses both scale and translational invariance, might be a better metric for measuring relationships in ILKDG.

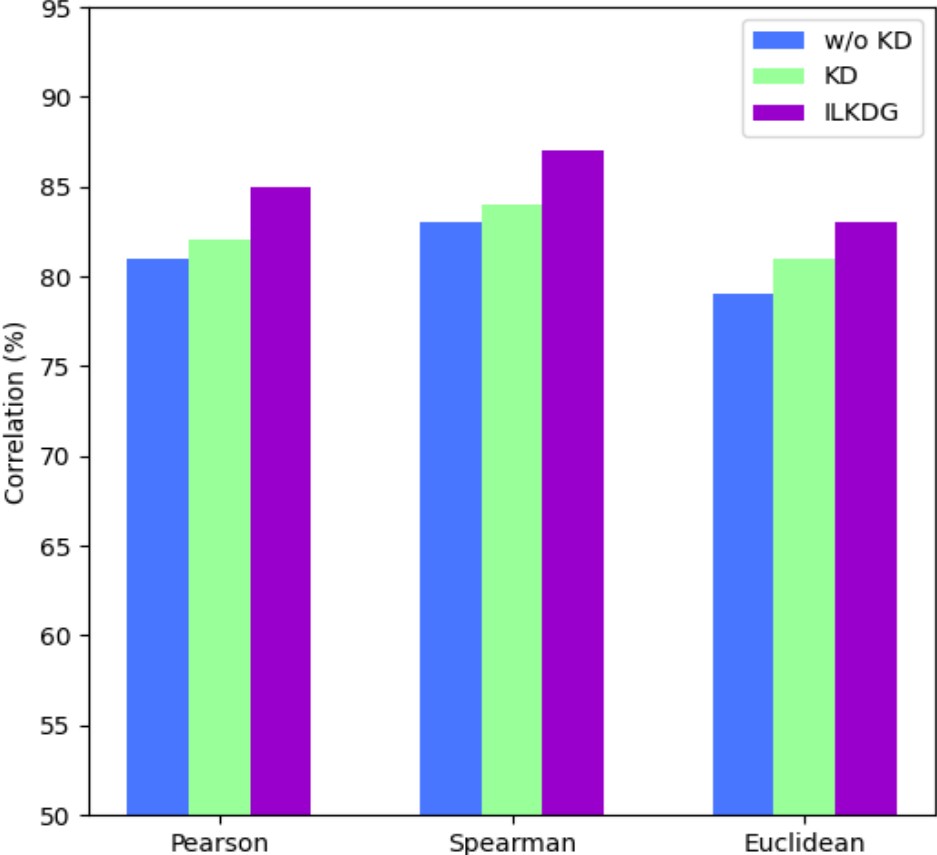

**Figure 7.** Correlations between student and teacher.

**Table 5.** Ablation of cosine similarity and Pearson correlation in ILKDG.

| Method | w/o LS | w/LS |
|---|---|---|
| Teacher | 0.892 | 0.895 |
| KD ($\tau = 6$) | 0.850 | 0.852 |
| KD ($\tau = 1$) | 0.853 | 0.859 |
| ILKDG (cosine) | 0.861 | 0.867 |
| ILKDG (Pearson) | 0.868 | 0.870 |

## 5. Conclusions and Future Research

This article introduces a novel knowledge distillation method called ILKDG and explores its integration with YOLOv7 in the context of forest fire detection applications. YOLOv7 is renowned for its efficient real-time detection capabilities but its performance may be constrained in resource-constrained environments. To address this challenge, we incorporated ILKDG into the YOLOv7 model, enabling knowledge transfer from a teacher model to a lighter-weight student model. This approach maintains high detection accuracy while reducing computational and memory requirements. We compared our ILKDG method with classic knowledge distillation algorithms like KD and current mainstream distillation methods, conducting experiments on a self-created forest fire detection dataset. The results demonstrate a significant improvement in detection performance due to the added benefit of ILKDG's distillation technique. Specifically, without changing student network parameters or network depth, we achieved a 2.9% improvement in mAP@0.5 and a 2.7% improvement in mAP@0.5:0.95. This indicates that combining YOLOv7 with the

ILKDG method proposed in this article effectively enhances the accuracy and performance of forest fire detection.

While this paper has already validated the effectiveness of the ILKDG method in enhancing the accuracy and performance of forest fire detection, there is still potential for further enhancement. Future research directions include the following four aspects. (1) Despite the proven effectiveness of ILKDG, significant potential exists for further improvement. Future research will optimize the ILKDG parameters and structure to enhance detection performance, including increasing mean average precision (mAP) and reducing false positive rates. (2) Additional techniques to achieve a lightweight model will be explored to adapt to resource-constrained environments. This may involve investigating compact network designs, quantization techniques, or model pruning, building upon ILKDG's existing capacity to reduce computational and memory requirements. (3) Further refinement of the dataset and data augmentation techniques are crucial to enhancing the model's robustness. Incorporating diverse data, including different weather conditions, periods, geographical locations, and fire incidents, will improve the model's overall performance in real-world scenarios. (4) Another intriguing research direction involves exploring semi-supervised learning techniques. Integrating data from different sensors, such as infrared images and smoke sensor data, alongside visual data could ensure accurate detection results under diverse weather conditions, ultimately enhancing the precision and reliability of fire detection.

**Author Contributions:** J.X.: Methodology, writing—original draft, conceptualisation, validation, project administration. H.Z.: Visualisation, data acquisition, investigation, software. All authors have read and agreed to the published version of the manuscript.

**Funding:** This work was supported by the Natural Science Foundation of China (Grant No. 62276276).

**Institutional Review Board Statement:** The study did not involve humans or animals.

**Informed Consent Statement:** The study did not involve humans or animals.

**Data Availability Statement:** The data presented in this study are available on request from the corresponding author.

**Acknowledgments:** We sincerely thank the anonymous reviewers for their critical comments and suggestions for improving the manuscript.

**Conflicts of Interest:** The authors declare no conflict of interest.

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
