# Peer review of "Forest Fire Object Detection Analysis Based on Knowledge Distillation"

_fire, doi:10.3390/fire6120446_

Round 1

Reviewer 1 Report

Comments and Suggestions for Authors

SEEM

This paper presents a new knowledge distillation method called ILKDG and explores its integration with YOLOv7 in the context of wildfire detection applications. The Article presents a new development with technical-scientific potential to be published in the Journal Fire/MDPI. The methodological development and results presented are positive. However, in my opinion, the article needs some adjustments to be fully accepted.

Below are some points:

1) THE TITLE

The TITLE needs to be more scientific and representative of the article.

Suggestion: “Forest fire object detection analysis based on knowledge distillation”

2) ABSTRACT

It is necessary to present a concluding paragraph on the potential of the methodological development proposed in future studies.

3) THE INTRODUCTION

The INTRODUCTION needs to have a few more bibliographic citations related to the main theme of the article. The article presents a total of only 44 references. Remove Figure 1 from the INTRODUCTION. The article is not a book chapter. Item 2.1 must be inserted in MATERIALS AND METHODS. If you can synthesize it, it will be better.

4) CONCLUSIONS

It is necessary to present a concluding paragraph on the potential of the methodological development proposed in future studies.

5) REFERENCES

The article presents a total of only 44 references. I consider this number small. See if you can reach the number of 60 bibliographic references

Reviewer 2 Report

Comments and Suggestions for Authors

The authors have made interesting attempts at Forest Fire Object Detection in a new approach.

1 There are few papers citing this journal in the references, and it is difficult to see the relevance and continuity of this paper with this journal. Authors are advised to cite the journal papers appropriately and summarize them.

2 The description of the methods section of the paper should be supplemented with a description of what is the result of citing other people's papers and what is the author's innovative content

3 The conclusion should be more structured, such as using (1), (2)...

4 Part of the English grammar problems.

Comments on the Quality of English Language

Some English grammar problems should be revised.
